# GRACE-C: Generalized Rate Agnostic Causal Estimation via Constraints

**Mohammadsajad Abavisani***
Department of Electrical and Computer Engineering
Georgia Institute of Technology
Atlanta, GA 30332
`s.abavisani@gatech.edu`

**David Danks**
Halicioglu Data Science Institute and
Department of Philosophy
University of California San Diego
San Diego, CA 92093
`ddanks@ucsd.edu`

**Sergey M. Plis**
TReNDS center
Department of Computer Science
Georgia State University
Atlanta, GA 30302
`s.m.plis@gmail.com`

## Abstract

Graphical structures estimated by causal learning algorithms from time series data can provide misleading causal information if the causal timescale of the generating process fails to match the measurement timescale of the data. Existing algorithms provide limited resources to respond to this challenge, and so researchers must either use models that they know are likely misleading, or else forego causal learning entirely. Existing methods face up-to-four distinct shortfalls, as they might *a*) require that the difference between causal and measurement timescales is known; *b*) only handle very small number of random variables when the timescale difference is unknown; *c*) only apply to pairs of variables; or *d*) be unable to find a solution given statistical noise in the data. This paper addresses these challenges. Our approach combines constraint programming with both theoretical insights into the problem structure and prior information about admissible causal interactions to achieve multiple orders of magnitude in speed-up. The resulting system maintains theoretical guarantees while scaling to significantly larger sets of random variables ($> 100$) without knowledge of timescale differences. This method is also robust to edge misidentification and can use parametric connection strengths, while optionally finding the optimal solution among many possible ones.

## 1 Introduction

Dynamic causal models play a pivotal role in modeling real-world systems in diverse domains, including economics, education, climatology, and neuroscience. Given a sufficiently accurate causal graph over random variables, one can predict, explain, and potentially control some system; more generally, one can understand it. In practice, however, specifying or learning an accurate causal model of a dynamical system can be challenging for both statistical and theoretical reasons.

One particular challenge arises when data are not measured at the speed of the underlying causal connections. For example, fMRI scanning of the brain indirectly measures dynamical neural activity by measuring the resulting bloodflow and oxygen level changes in different brain regions. However, fMRI measures occur (at most) every second while the brain's actual dynamics are known to proceed at a faster rate (Oram & Perrett, 1992), though we do not know how much faster. In general, when the measurement timescale is significantly slower than the causal timescale (as with fMRI), learning can output vastly incorrect causal information. For instance, if we only measure every other timestep

---

*Corresponding author

in Figure 1, then the true graph (top left) would differ from the data graph (top right). We might thus conclude that variable 2 directly influences variable 5, when variable 3 is the actual direct cause. These errors can lead to inefficient or costly attempts at control. More generally, understanding of a system depends on the timescale of the causal relations, not the timescale of measurements.

In this paper, we consider the problem of learning the causal structure at the *causal* timescale from data collected at an unknown *measurement* timescale. This challenge has received significant attention in recent years (Plis et al., 2015b; Gong et al., 2015; Hyttinen et al., 2017; Plis et al., 2015a), but all current algorithms have significant limitations (see Section 2) that make them unusable for many real-world scientific challenges. Current algorithms show the theoretical possibility of causal learning from undersampled data, but their practical applicability is limited to small graph sizes, perhaps only a pair of variables (Gong et al., 2015). In contrast, we present a provably correct and complete solution that can operate on 100-node graphs, and hence is potentially applicable in biological and other domains, for learning causal timescale structure from undersampled data.

## 2 RELATED WORK AND NOTATION

A directed dynamic causal model is a generalization of "regular" causal models (Pearl et al., 2000; Spirtes et al., 1993): graph $\mathbf{G}$ includes $n$ distinct nodes for random variables $\mathbf{V} = \{V_1, V_2, ..., V_n\}$ at both the current timestep $t$ ($\mathbf{V}^t$), and also previous timesteps ($\mathbf{V}^{t-k}$) for which there is a direct cause of some $V_i^t$. We assume that the "true" underlying causal structure is first-order Markov: the independence $\mathbf{V}^t \perp\!\!\!\perp \mathbf{V}^{t-\tilde{k}} \mid \mathbf{V}^{t-1}$ holds for all $k > 1$.[1] $\mathbf{G}$ is thus over 2$\mathbf{V}$, and the only permissible edges are $V_i^{t-1} \rightarrow V_j^t$, where possibly $i = j$. The quantitative component of the dynamic causal model is fully specified by parameters for $P(\mathbf{V}^t|\mathbf{V}^{t-1})$. We assume that these conditional probabilities are stationary over time, but the marginal $P(\mathbf{V}^t)$ need not be stationary.

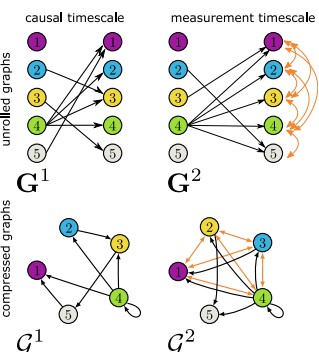

Figure 1: Causal graph $\mathbf{G}^1$ and its undersampled version $\mathbf{G}^2$: unrolled and compressed versions.

We denote the timepoints of the underlying causal structure as $\{t^0, t^1, t^2, ..., t^k, ...\}$. The data are said to be *undersampled at rate* $u$ if measurements occur at $\{t^0, t^u, t^{2u}, ..., t^{ku}, ...\}$. We denote undersample rate with superscripts: the true causal graph (i.e., undersampled at rate 1) is $\mathbf{G}^1$; the same graph undersampled at rate $u$ is $\mathbf{G}^u$. To determine the implied $\mathbf{G}$ for $u > 1$, the graph is first "unrolled" by adding instantiations of $\mathbf{G}^1$ at previous timesteps, where $\mathbf{V}^{t-2}$ bear the same causal relationships to $\mathbf{V}^{t-1}$ that $\mathbf{V}^{t-1}$ bear to $\mathbf{V}^t$, and so forth. In this unrolled (time-indexed by $t$) graph, all $\mathbf{V}$ at intermediate timesteps are unmeasured; this lack of measurement is equivalent to marginalizing out (the variables in) those timesteps to yield $\mathbf{G}^u$. The problem of moving from $\mathbf{G}^1$ to $\mathbf{G}^u$ was structurally addressed by Danks & Plis (2013), and parametrically addressed (for 2-variable systems) by Gong et al. (2015).

Various representations have been developed for graphs with latent confounders, including partially-observed ancestral graphs (PAGs) (Zhang, 2008) and maximal ancestral graphs (MAGs) (Richardson & Spirtes, 2002). However, these graph-types cannot easily capture the types of latents produced by undersampling (Mooij & Claassen, 2020). Instead, we use compressed graphs, along with properties that were previously proven for this representation (Danks & Plis, 2013). A compressed graph includes only $\mathbf{V}$, where temporal information is implicitly encoded in the edges. In particular, a compressed graph $\mathcal{G}$ for dynamic causal graph $\mathbf{G}$ has $V_i \rightarrow V_j$ in $\mathcal{G}$ iff $V_i^{t-1} \rightarrow V_j^t$ is in $\mathbf{G}$. Undersampling (i.e., marginalizing intermediate timesteps) is a straightforward operation for compressed graphs: (1) $V_i \rightarrow V_j$ in $\mathcal{G}^u$ iff there is a length-$u$ directed path from $V_i$ to $V_j$ in $\mathcal{G}^1$ iff there is a directed path from $V_i^{t-u}$ to $V_j^t$ in $\mathbf{G}^1$; and (2) $V_i \leftrightarrow V_j$ in $\mathcal{G}^u$ iff there exists length-$s < u$ directed paths from $V_k$ to $V_i$, and to $V_j$, in $\mathcal{G}^1$ (i.e., $V_k$ is an unobserved common cause in $\mathbf{G}^1$ fewer than $u$ timesteps back). (See Appendix A for additional lemmas and proofs.) The bottom row of Figure 1 shows compressed graphs for the unrolled ones on the top row; the left shows the causal timescale and the right shows the graphs undersampled at rate 2. (See Appendix B for more examples of graphs through undersampling.)

---

[1]This assumption is relatively weak, as we do not assume that we measure at this causal timescale. The causal timescale could be arbitrarily fast. This assumption is a form of causal sufficiency (Spirtes et al., 2000).

Given this framework, the overall causal learning challenge can now be stated as: given $\mathcal{G}^u$ but not $u$ (alternately: given dataset **D** at unknown undersample rate), what is the set of possible $\mathcal{G}^1$? There will often be many possible $\mathcal{G}^1$ that appear the same after undersampling, and so we use $[\![\mathcal{H}]\!]$ to denote the equivalence class of $\mathcal{G}^1$ that yield $\mathcal{H}$ (the given causal graph inferred from data **D**) for some $u$. That is, $[\![\mathcal{H}]\!] = \{\mathcal{G}^1 : \exists u(\mathcal{G}^u = \mathcal{H})\}$. There are $2^{n^2}$ possible $\mathcal{G}^1$, so perhaps unsurprisingly, the problem of inferring $[\![\mathcal{H}]\!]$ is NP-complete:

**Theorem 1** (Hyttinen et al. (2017)[Theorem 1]). *Deciding whether a consistent $\boldsymbol{G}^1$ exists for a given $\mathcal{H}$ is NP-complete, for all undersampling rates $u \geq 2$.*[2]

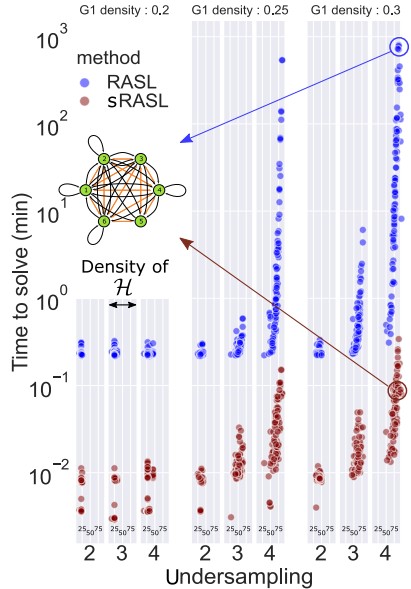

Figure 2: Comparison of sRASL (red) with previous state-of-the-art RASL (blue).

Several algorithms exist for this problem. *Mesochronal Structure Learning (MSL)* (Plis et al., 2015b) infers $[\![\mathcal{H}]\!]$ in a non-brute force manner given known $u$. Every edge in $\mathbf{G}^u$ corresponds to one or more paths of length $u$ in $\mathbf{G}^1$, and so $\mathbf{G}^1$ can be constructed by identifying $u - 1$ intermediate nodes for each edge in $\mathbf{G}^u$. MSL uses Depth-First Search (DFS) through the state space of possible identifications, where each implies a $\mathbf{G}^1$. If $\mathbf{G}^u = \mathcal{H}$, then $\mathbf{G}^1 \in [\![\mathcal{H}]\!]$. Otherwise, search continues. MSL backtracks in the DFS whenever some $\mathbf{G}^u$ includes an edge that is absent from $\mathcal{H}$, as the candidate $\mathbf{G}^1$ and all its supergraphs cannot be in $[\![\mathcal{H}]\!]$.

Although Plis et al. (2015b) showed that the concept of causal inference from undersampled data is feasible, MSL is computationally intractable on even moderate-sized graphs. Hyttinen et al. (2017) used the implied constraints to develop an Answer Set Programming (ASP) (Simons et al., 2002; Niemelä, 1999; Gelfond & Lifschitz; Lifschitz, 1988) method that formulated this causal inference challenge as a rule-based constraint satisfaction problem that is well-suited for ASP-type solvers. In essence, the algorithm in Hyttinen et al. (2017) takes as input the measured causal graph $\mathcal{H}$, determines the set of implied constraints on $\mathbf{G}^1$, and then uses the general-purpose Answer Set Solver *Clingo* (Gebser et al., 2011) to determine the set of possible $\mathbf{G}^1$ significantly faster than MSL. The same idea of using Boolean satisfiability solvers to integrate (in)dependent data constraints has been used for various other causal learning challenges (Hyttinen et al., 2013; Triantafillou et al., 2010).

Although the method in Hyttinen et al. (2017) is significantly faster than MSL, one must specify the undersampling rate $u$ (or else run the method sequentially for all possible $u$, thereby losing much of the computational advantage). In contrast, the *Rate-Agnostic (Causal) Structure Learning (RASL)* approach (with several variants) (Plis et al., 2015a) makes no such assumption. RASL algorithms are similar to MSL, but consider each possible $u$ for some $\mathbf{G}^1$. RASL reduces computational complexity with two additional stopping rules for given $\mathbf{G}^1$: (1) if some $\mathbf{G}^k$ has previously been seen, then further undersampling of $\mathbf{G}^1$ will not produce new graphs; and (2) if $\mathbf{G}^k$ is not an edge-subset of $\mathcal{H}$ for all $k$, then do not consider any edge-superset of $\mathbf{G}^1$ (Plis et al., 2015a). Despite these improvements, RASL still faces memory and run-time constraints for even moderate numbers of nodes.

One key observation from all of these learning algorithms is the importance of *strongly connected components* (SCCs) (Danks & Plis, 2013), where the variables in a compressed graph $\mathcal{H}$ can be fully partitioned based on SCC membership.

**Definition 2.1.** *An SCC in compressed graph $\mathcal{H}$ is a maximal set of nodes $S \subseteq V$ such that, for every $X, Y \in S$ there is a directed path from $X$ to $Y$.*

SCCs can be highly stable despite undersampling: the node-membership of an SCC does not change as we undersample, as long as the greatest common divisor (gcd) of the set of lengths of all simple loops (directed cycles without repeated nodes) in the SCC is 1.[3]

---

[2]Proof provided in Hyttinen et al. (2017). In general, we omit previously published proofs.

[3]The condition easily holds, as it requires only (1) the graph is relatively dense with different loop lengths, or (2) at least one node in the SCC has a self-loop (i.e., is autocorrelated).

**Theorem 2** (Danks & Plis (2013)[Theorem 3]). *$S$ is an SCC in $G^u$ $\forall u$ iff $gcd(\mathcal{L}_S) = 1$ for SCC $S \in G$*[1]

The algorithms in this paper all take as input the measurement timescale graph $\mathcal{H}$, perhaps estimated from data at an (unknown) undersampling rate. We do not here develop algorithms to learn $\mathcal{H}$, as there are many existing algorithms for learning graphical structure: at the measurement timescale (Chu et al., 2008; Entner & Hoyer, 2010; Granger, 1969); for time series with latent confounders (Jabbari et al., 2017; Malinsky & Spirtes, 2018; Gerhardus & Runge, 2020); or accounting for structured latents such as those that occur in undersampling (Moneta et al., 2011; Cook et al., 2017).

In this paper, we develop *sRASL* (for *s*olver-based RASL), a novel solution to the rate-agnostic structure learning problem that leverages multiple types of insights and constraints (e.g., Theorem 2), and thereby significantly outperforms previous methods. The contributions of this paper are threefold: first, we reformulate the RASL algorithm from a search-based procedure to a constraint satisfaction problem encoded in a declarative language (Fahland et al., 2009). Second, we show how to add additional constraints based on SCC structure. Third, we ensure that sRASL provides a straight-forward way to find approximate solutions when $\mathcal{H}$ is an unreachable graph (i.e., when $[\![\mathcal{H}]\!] = \emptyset$). These advances collectively provide up to three orders of magnitude improvements in speed, thereby enabling causal inference given undersampling data involving over 100 nodes. Figure 2 compares sRASL (red) with the previously-fastest RASL (Plis et al., 2015a) method (blue) on the same graphs. For the example $\mathcal{H}$, RASL required nearly 1000 minutes to compute $[\![\mathcal{H}]\!]$, but only 6 seconds for sRASL. Even the longest-to-compute $[\![\mathcal{H}]\!]$ for sRASL took 20.5 seconds vs. 780 minutes for RASL.

## 3 sRASL: Optimized ASP-based Causal Discovery

The sRASL approach takes as input a (potentially) undersampled graph $\mathcal{H}$, whether learned from data **D**, expert domain knowledge, both of these, or some other source. sRASL's agnosticism about the source of the input graph enables wider applicability, as we can use whatever information is available (Danks & Plis, 2019). In the asymptotic (data) limit, the sRASL output is the full $[\![\mathcal{H}]\!]$.

sRASL leverages the fact that connections *between* SCCs in $\mathcal{H}$ must form a directed acyclic graph. More specifically: if $X \rightarrow Y$ with $X \in \mathbf{A}, Y \in \mathbf{B}$ for SCCs $\mathbf{A} \neq \mathbf{B}$, then $C \leftrightarrow D$ for all $C \in \mathbf{A}, D \in \mathbf{B}$.[4] Theorem 2 provides the (weak) condition under which SCC membership is preserved under undersampling. These two observations imply that structural features provide additional constraints beyond the obvious ones (see Section 4.3). In particular, if $\mathcal{H}$ has a roughly modular structure, then sRASL generates many more constraints than the formulation of Hyttinen et al. (2017). (See Appendix D for an ablation study of speed effects of these added constraints.)

Listing 1 shows the `Clingo` (see Appendix F for a brief introduction) code[5] of sRASL, which involves exactly representing the conditioning and marginalization operations (from Section 2) in ASP. Line 1 specifies the first-order graph structure of $\mathcal{H}$ (e.g., the edge $1 \rightarrow 10$ translates to hdirected(1, 10)). Line 2 encodes the second-order structure of $\mathcal{H}$, including the partition of **V** into SCCs. Separate code adds these predicates and basic descriptive information (Lines 3, 4, 5) to the `Clingo` code. `maxu` on line 3 specifies the maximum undersampling rate; noter that there is provably a $u$ where $G^u = G^k$ for all $k > u$, under the same condition that leads to stable SCC membership:

**Theorem 3** (Plis et al. (2015a)[Theorem 3.1]). *If $gcd(\mathcal{L}_S) = 1$ for all SCCs $S \subseteq V$, then $G^u = G^{u+1}$ for all $u > f \leq n_F + \gamma + d + 1$.*

where $\gamma$ is the transit number[6], $d$ is graph diameter[7], and $n_F$ is the Frobenius number.[8] In practice, the plausible undersampling rate will often be much lower than the theoretical upper bound in Theorem 3, and so `maxu` could be set by expert knowledge. For example, the underlying rate of brain activity is generally thought to be $\sim$ 100 milliseconds and fMRI devices measure approximately every two seconds. Hence, $u = 20$ is a plausible upper bound on undersampling in fMRI studies.[9]

---

[4]If $C \leftrightarrow D$, then by definition of SCC, there exists $\pi : X \leftarrow \ldots \leftarrow C \leftarrow D \leftarrow \ldots \leftarrow Y$. $X, Y$ are thus mutually reachable, so must be in the same SCC, contra $\mathbf{A} \neq \mathbf{B}$.

[5]The full code is available at https://gitlab.com/undersampling/gunfolds

[6]Length of the "longest shortest path" from a node that touches all simple loops of the SCC.

[7]Length of the "longest shortest path" between any two graph nodes.

[8]For set **B** of positive integers with gcd(**B**) = 1, $n_F$ is the max integer with $n_F \neq \sum_{i=1}^{b} \alpha_i B_i$ for $\alpha_i \geq 0$

[9]Of course, the actual undersample rate could be much lower than 20. Voxels typically contain $8 - 10$ layers of neurons, so the "causal timescale of a voxel" could easily be as high as 1000 ms (i.e., $u = 2$).

```
1  %( * input graph edge specifications here * e.g.: hdirected(1,5) ... )
2  %( * input graph SCC specifications here * e.g.: sccsize(0, 5). scc(1, 0) ...)
3  #const n = 10, maxu = 20
4  node(1..n).
5  1 {u(1..maxu)} 1.
6  {edge1(X,Y)} :- node(X), node(Y).
7  directed(X, Y, 1) :- edge1(X, Y).
8  directed(X, Y, L) :- directed(X, Z, L-1), edge1(Z, Y), L <= U, u(U).
9  bidirected(X, Y, U) :- directed(Z, X, L), directed(Z, Y, L), node(X;Y;Z), X < Y, L
       < U, u(U).
10 :- directed(X, Y, L), not hdirected(X, Y), node(X;Y), u(L).
11 :- bidirected(X, Y, L), not hbidirected(X, Y), node(X;Y), u(L), X < Y.
12 :- not directed(X, Y, L), hdirected(X, Y), node(X;Y), u(L).
13 :- not bidirected(X, Y, L), hbidirected(X, Y), node(X;Y), u(L), X < Y.
14 % the following is only used when SCC accounting is enabled
15 :- edge1(X, Y), scc(X, K), scc(Y, L), K != L, sccsize(L, Z), Z > 1, not dag(K,L).
```

Listing 1: sRASL encoding in the `clingo` ASP-language

```
1      :~ directed(X, Y, L), no_hdirected(X, Y, W), node(X;Y), u(L). [W@1,X,Y]
2      :~ bidirected(X, Y, L), no_hbidirected(X, Y, W), node(X;Y), u(L), X < Y.
          [W@1,X,Y]
3      :~ not directed(X, Y, L), hdirected(X, Y, W), node(X;Y), u(L). [W@1,X,Y]
4      :~ not bidirected(X, Y, L), hbidirected(X, Y, W), node(X;Y), u(L), X < Y.
          [W@1,X,Y]
```

Listing 2: Integrity constraints to replace Lines 11-14 in Listing 1 to convert sRASL into optimization problem

Line 6 in Listing 1 stipulates that all edges in $\mathcal{G}^1$ are possible (by default), and so the output will contain any possible model that does not violate the integrity constraints of lines $11 - 15$. Lines 7 and 8 define paths of length $L$ in the graph (i.e., an edge in $\mathcal{G}^L$). As described in Section 2, $X \to Y \in \mathcal{G}^u \iff X \overset{u}{\leadsto} Y \in \mathcal{G}^1$ where $\overset{u}{\leadsto}$ is a path of length $u$. Line 9 similarly defines bidirected edges in $\mathcal{G}^L$: $X \leftrightarrow Y \in \mathcal{G}^u \iff \exists Z, l : (X \overset{l}{\leftsquigarrow} Z \overset{l}{\leadsto} Y \in \mathcal{G}^1)$.

Lines $10 - 13$ provide the core constraints that ensure sRASL returns only $\mathcal{G}^1$ for which there exists $u$ with $\mathcal{G}^u = \mathcal{H}$. Line 15 adds the constraint of impermissibility of cycles between SCCs: if each SCC is considered as a *super-node*, Line 15 ensures that the graph of SCC connections in $\mathcal{H}$ is acyclic.

sRASL can return the empty set (i.e., there are no suitable $\mathcal{G}^1$), typically because of statistical noise or other errors in estimating or specifying $\mathcal{H}$.[10] We can instead run sRASL in an optimization mode to find optimal (though not perfect) outputs (see Section 4.5 for details). In such cases, sRASL finds the set of $\mathbf{G}^1$ that are, for some $u$, closest to $\mathcal{H}$ by the objective function:

$$\mathcal{G}^{1*}, u^* \in \arg\min \sum_{e \in \mathcal{H}} I[e \notin \mathcal{G}^u] \cdot w(e \in \mathcal{H}) + \sum_{e \notin \mathcal{H}} I[e \in \mathcal{G}^u] \cdot w(e \notin \mathcal{H}), \tag{1}$$

where the indicator function $I(c) = 1$ if $c$ holds and 0 otherwise. $w(e \in \mathcal{H})$ indicates the importance (i.e., reliability) of edge $e$; $w(e \notin \mathcal{H})$ indicates the reliability of the absence of an edge. Since $\mathcal{H}$ is an undersampled graph, it consists of directed and bidirected edges. We thus implement both $w(e \in \mathcal{H})$ and $w(e \notin \mathcal{H})$ as two pairs of $n \times n$ matrices, one pair for existence and absence of directed edges, and one pair for bidirected edges. To learn the optimal graph at the true causal timescale, the corresponding $\mathcal{G}^u$ of each $\mathcal{G}^1$ in the solution set is compared to $\mathcal{H}$, and penalized for the difference according to the weights.

The reliability weights may also be based on strength of connection. For example, if $\mathcal{H}$ is estimated as a Granger Causality or Structural Vector Autoregressive (SVAR) (Granger, 1969; Lütkepohl, 2005) model, then the edge-weights may enable `Clingo` to preferentially ignore edges with weaker

---

[10]Among all possible graphs that have a combination of both directed ($2^{n^2}$) and bidirected ($2^{\binom{n}{2}}$) edges, only a fraction are possible by undersampling a $\mathcal{G}^1$.

connection strength. In addition to using observed data to estimate the weights, prior knowledge can play a key role: edges known to exist can be given a higher weight, while those known to not exist could be given reduced (or zero) weight (See also Appendix D). The approach is flexible as it can combine estimates from data and expert knowledge. Applications to fMRI data for causal structure discovery at causal time scale are shown in Appendix E.

In order to incorporate Equation 1 in Listing 1, we replace its exact integrity constraints (Lines $10 - 13$) with the optimization formulation (Gebser et al., 2011) in Listing 2. In Listing 2, we specify a weight for each edge (or lack there of) in $\mathcal{H}$ using $\mathtt{W}$, with importance specified using $\mathtt{W@i}$ syntax with $\mathtt{i}$ being the importance.

### 3.1 sRASL COMPLETENESS AND CORRECTNESS

sRASL exhibits significant improvements in computation time, so it is important to show that we do not lose generality or theoretical guarantees. We demonstrate correctness and completeness using the notion of a *direct encoding* of the problem (i.e., the space of solutions is fully characterized, and any non-solution violates a constraint). We first prove (see Appendix A):

**Theorem 4.** *Listing 1 is a direct encoding of the undersampling problem.*

$\mathtt{Clingo}$ is a complete solver, based on CDNL (Conflict-Driven Nogood Learning) (Drescher & Walsh, 2011), itself based on CDCL (Conflict-Driven Clause Learning) (Marques Silva & Sakallah, 1996; Marques-Silva & Sakallah, 1999). Hyttinen et al. (2014)[Theorem 2] and Hyttinen et al. (2013)[Section 5.2] show that, if the ASP encoding is the direct encoding of the problem, then ASP will produce the complete set of solutions in the infinite sample space limit. In other words, Theorem 3.1 implies: since our algorithm yields at least one sound solution, $\mathtt{Clingo}$ will produce all possible solutions. Therefore, soundness results in completeness. That is, sRASL's success is not due to heuristics or some incomplete or not-everywhere-correct algorithmic step.[11]

## 4 RESULTS

A major virtue of sRASL is its empirical performance, so we now consider a range of simulations (where we have known ground truth) to understand this performance in more detail. These experiments used $\mathtt{Clingo}$ in parallel mode using 10 threads computing on $\mathtt{AMD\ EPYC\ 7551}$ CPUs. Given computational complexity, all experiments were run on a $\mathtt{slurm}$ cluster that submits jobs to one of the 19 machines on the same network, each with 64 cores and 512 GB of RAM.

### 4.1 COMPARING sRASL VS. RASL

We first compare sRASL with the existing RASL method, which struggles with graphs larger than 6 nodes (Plis et al., 2015a) (Figure 2). We generated 100 6-node SCCs for each density in $[0.2, 0.25, 0.3]$, and then undersampled each graph by 2, 3, and 4. Each column of Figure 2 consists of graphs of approximately same density (increasing density from left-to-right), and subcolumns represent different undersample rates (for that density). As Figure 2 shows, sRASL is typically three orders of magnitude faster than RASL, even on small graphs. A similar comparison with the method of Hyttinen et al. (2017) that iteratively loops through possible values of $\mathtt{u}$ can be found in Figure 9 of Appendix C.

### 4.2 COMPARING GRAPH SIZE

It is perhaps unsurprising that sRASL runs much faster than RASL, as sRASL uses an ASP solver (which were previously known to yield faster algorithms (Hyttinen et al., 2017)). We next explored how large graphs could be that sRASL could handle. More generally, we aimed to better understand how sRASL's computational performance scales with the number of nodes in single-SCC graphs. The focus on single SCCs is motivated by the theoretical need to understand the size-speed tradeoff. Moreover, many real-world systems consist of tightly coupled factors with many feedback loops (i.e., a single SCC). We consider multiple-SCC graphs in the next subsections.

---

[11]Simulation testing provides further evidence. We found that sRASL and RASL produced identical outputs for 1000 different input graphs, and RASL is known to be correct and complete (Plis et al., 2015a)[Theorem 3.6].

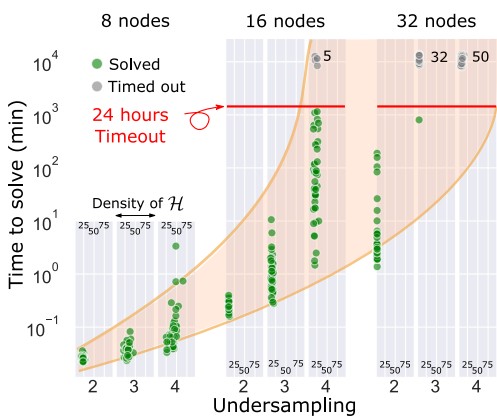

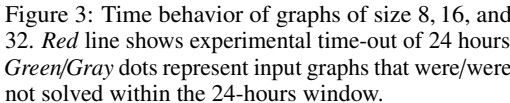

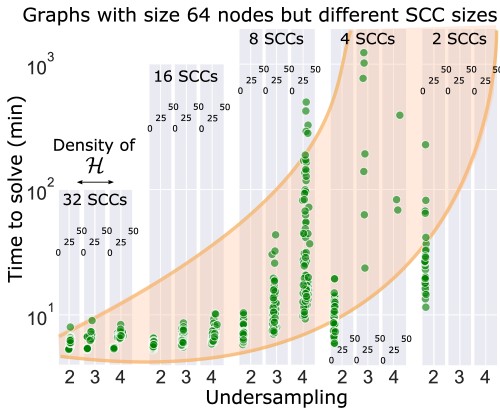

Figure 3: Time behavior of graphs of size 8, 16, and 32. *Red* line shows experimental time-out of 24 hours. *Green/Gray* dots represent input graphs that were/were not solved within the 24-hours window.

Figure 4: Time behavior of graphs of size 64 with various SCC sizes. The time-out for this experiment was 24 hours (1440 Minutes).

We generated 50 random single-SCC graphs each of 8, 16, and 32 nodes, all with average degree of 1.4 outgoing edges per node. We then undersampled each graph by 2, 3, and 4, and used each individual undersampled graph as input to sRASL (i.e., 150 different input graphs for each size). We used a 24-hour timeout (i.e., stopped the run if it did not finish in 24 hours). Figure 3 shows the increasing computational costs as both number of nodes and undersample rate increase. Notably, sRASL was able to learn ⟦$\mathcal{H}$⟧ for many 32-node single-SCC graphs, though it reached timeout for all 32-node $\mathcal{H}$ at $u = 4$. That is, for low $u$, sRASL scales to much larger single-SCC graphs than RASL.

### 4.3 Comparing SCC Size in Multiple-SCC Graphs

The other major innovation of sRASL is incorporation of constraints derived from the SCC structure. We thus investigated the performance of sRASL on large, structured, multiple-SCC graphs. Many real-world systems exhibit some degree of modularity, where there are dense or feedback connections within a module or subsystem, and relatively sparser connections between modules or subsystems. In theory, sRASL should perform well on these kinds of structures since it incorporates SCC-based constraints. Please refer to Appendix D for an ablation study on the marginal benefit provided by these additional constraints for SCC structures.

We tested the value of SCC-based constraints using graphs with 64 nodes that differed in their SCC structure. Specifically, we randomly generated 50 graphs each of: 32 size-2 SCCs; 16 size-4 SCCs; 8 size-8 SCCs; 4 size-16 SCCs; or 2 size-32 SCCs. We then undersampled each graph by $u = 2, 3$, or 4, and ran sRASL (again with a 24-hour timeout).

Figure 4 shows the computation time for these graphs, with increasing SCC size (and decreasing number of SCCs) from left to right. The first key observation is that sRASL successfully found ⟦$\mathcal{H}$⟧ for 64-node graphs, at least when there was some internal structure. Second, and relatedly, we observe a wide range of computation times for these graphs, even though all had the same number of nodes (64). We clearly see the impact of SCC structure, as sRASL was dramatically faster when there were many small SCCs, rather than a few large SCCs. The results in Figure 3 might seem to suggest an "upper bound" around 30 nodes for sRASL. But the results in Figure 4 make it clear that any potential "upper bound" is primarily on the number of nodes *within the SCCs*, rather than the total number of nodes in the graph.

### 4.4 Comparing Graph Size With Constant SCC Size

The previous results suggest that sRASL might be able to solve much larger graphs, as long as the SCCs are not overly large. More generally, the previous simulations showed that sRASL's computational cost scales (at least) exponentially in the *size* of the SCC, but did not reveal how it scales in the *number* of SCCs.

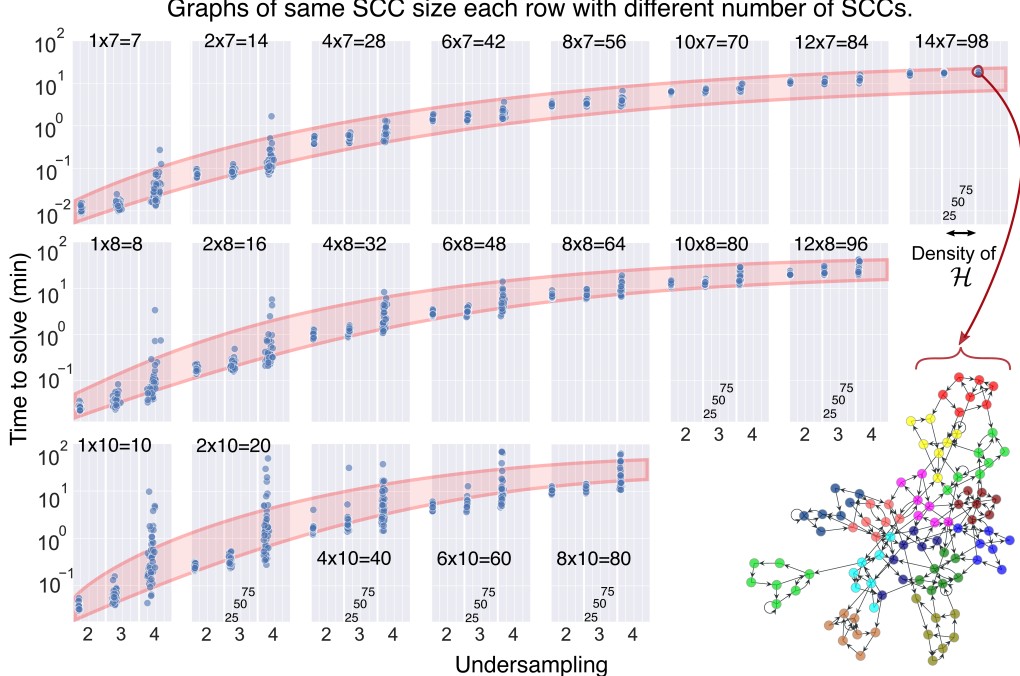

Figure 5: Time behaviour of graphs with the same SCCs sizes but with multiple number of SCCs. Top row: graphs of SCC size 7 with 1, 2, ..., 14 number of SCCs. Middle row: graphs of SCC size 8. Bottom row: graphs of SCC size 10. Bottom-right corner shows an example of a structured graph with 98 nodes composed of 14 SCCs of size 7. Each color represents one SCC.

We again generated 50 different graphs for each of several settings. We used SCCs with 7, 8, and 10 nodes, and varied the number of SCCs within a graph (again for $u = 2, 3$, and 4). Figure 5 shows the computational cost of sRASL, where each row includes graphs whose SCCs are the same size, and the number of SCCs increases from left-to-right. The critical observation here is that the time complexity grows approximately *linearly* in the number of SCCs, rather than exponentially (or worse). For example, the graph shown in Figure 5 has 98 nodes, but sRASL successfully computes $\llbracket \mathcal{H} \rrbracket$ in approximately 20 minutes. (Recall that RASL took 17 hours to compute a graph with only 6 nodes.)

This simulation demonstrates that sRASL is usable on relatively large graphs, as long as there is appropriate internal structure. One might worry, though, whether real-world systems have the right structure. For fMRI (brain) data, Sanchez-Romero et al. (2019) recently aggregated a number of simulations of realistic causal graphs for brain processes studied with fMRI, and the largest SCC in these widely-accepted models has only 7 nodes. Moreover, typical brain parcellations contain only $50 - 100$ regions (= nodes), and sRASL can easily handle 100-node graphs if SCCs are $8 - 10$ nodes.

The results in this subsection suggest that we could potentially find $\llbracket \mathcal{H} \rrbracket$ for even larger graphs ($n > 100$), as long as they were composed of reasonably-sized SCCs. However, we found that the `Clingo` language and solver seems to be limited in the number of atoms that it can handle. In our simulations, graphs of size 100 seem to be the limit for `Clingo` to handle all the predicates. An open question is whether sRASL can be optimized to produce fewer predicates (or `Clingo` improved to handle more atoms).

### 4.5 OPTIMIZATION

Finally, we explored the optimization capability of `Clingo`. Recall that sometimes $\llbracket \mathcal{H} \rrbracket = \emptyset$ due to statistical errors or other noise in learning $\mathcal{H}$. `Clingo` can solve an optimization problem based on user-specified weights and priorities, and output a single solution with minimum cost function (along with $u$ for this solution). In particular, we can use `Clingo` to find $\mathcal{G}^1$ whose $\mathcal{G}^u$ (for some $u$) are closest (relative to the edge weights) to $\mathcal{H}$.[12]

---

[12]If $\llbracket \mathcal{H} \rrbracket \neq \emptyset$, then this optimization will simply return a single graph from $\llbracket \mathcal{H} \rrbracket$.

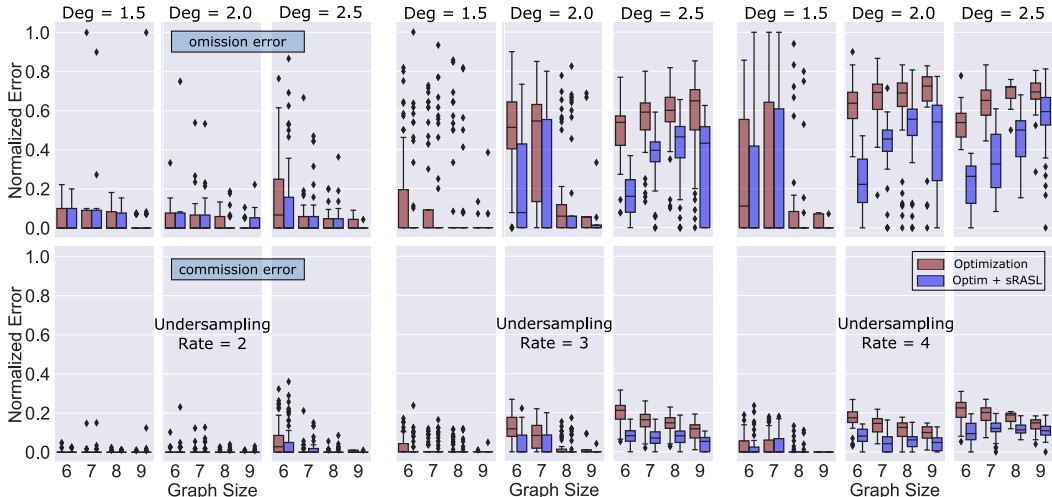

Figure 6: The omission (top) and commission (bottom) error of different graph sizes and undersampling of two, three and four from left to right.

In this simulation, we first randomly generate $\mathcal{G}^1$ and undersample at a random $u$ to get $\mathcal{G}^u = \mathcal{H}$ such that $[\![\mathcal{H}]\!] \neq \emptyset$. We then assign weights to the edges of $\mathcal{H}$ and randomly delete one edge in $\mathcal{H}$. We run sRASL on this "broken" $\mathcal{H}$ to learn a suitable $\mathcal{G}^1$. Red bars in Figure 6 show the edge omission and commission errors for this approach. We see that, except for high undersamplings, the optimization capability of Clingo can be used to frequently retrieve the true $\mathcal{G}^1$; that is, this version of sRASL is robust to small errors in $\mathcal{H}$ in many settings.

A more complex approach is to first run the optimization method to identify a solution $\mathbf{G}^1_{opt}$ and undersample rate $u_{opt}$. We can then undersample this solution $\mathbf{G}^1_{opt}$ by $u_{opt}$ to get $\mathbf{G}^u_{opt}$. We then use sRASL to obtain $[\![\mathbf{G}^u_{opt}]\!]$ (i.e., the full equivalence class of the undersampled graph that is "nearest" to $\mathcal{H}$). We then compute the error based on the minimum error among all $\mathbf{G}^1 \in [\![\mathbf{G}^u_{opt}]\!]$; that is, we ask whether the true graph was found *somewhere* in $[\![\mathbf{G}^u_{opt}]\!]$. This approach is motivated by the intended use of sRASL by domain scientists, where they can use domain knowledge to help select graph(s) from the equivalence class. Blue bars in Figure 6 show that this more complex method provides improved performance compared to regular optimization.

## 5 Conclusion and Discussion

Real-world scientific problems frequently involve measurement processes that operate at a different timescale than the causal structure of the system under study. As causal learning and analysis methods are increasingly used to address societal and policy challenges, it is increasingly critical that we use methods that reveal usable information (while also being clear when we *cannot* infer some information). Obviously, like any method, sRASL could yield information that is misused, but the aim here is to provide another useful tool in the scientists' and policy-makers' toolboxes. If measurements occur at a slower rate than the causal influences, then causal discovery from those undersampled data can yield highly misleading outputs. Multiple methods have been developed to infer aspects of the underlying causal structure from the undersampled data/graph. However, the assumptions or computational complexities of those algorithms make them unusable for most real-world challenges. In this paper, we have developed and tested sRASL, a novel approach that is less subject to those same limitations. More specifically, sRASL provides all consistent solutions (without knowledge of exact undersampling rate) for large (100-node) graphs in a usable amount of time. sRASL also shows reasonable robustness to statistical error in the estimated graph by finding the closest consistent solution. Future research will focus on application of sRASL to actual neuroimaging data, and extensions to situations with multiple measurement modalities.

## 6 ACKNOWLEDGMENTS

This work was supported by NIH R01MH129047 and in part by NSF 2112455, and NIH 2R01EB006841. We are grateful to Antti Hyttinen, Matti Järvisalo, and Frederick Eberhardt for discussions on clingo.

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

## A   APPENDIX

We start with proving some results used in conversion of the DBN structures to their compressed graph representations.

**Lemma 1.** *For all u, $G_u$ contains no directed edges between variables at the same time step.*

*Proof.* $vvu = 1$ holds by assumption for $G_1$. For $u > 1$, every directed edge corresponds to a directed path of length $u$ in $G_1$. Since all directed edges in $G_1$ are from $t - 1$ to $t$ (or more generally, from $t - k$ to $t - (k + 1)$), every directed path in $G_1$ is from an earlier time step to the current one. Hence, no directed edge in $G_u$ can be from $V_i^t$ to $V_j^t$. □

**Lemma 2.** *If the Markov order of $G_1$ is 1, then the Markov order of all $G_u$ is also 1 (relative to measurement at rate u).*

*Proof.* The Markov order of a dynamic causal graph is the smallest $m$ such that $\mathbf{V}^t$ is independent of $\mathbf{V}^{t-r}$ given $\mathbf{V}^{t-1}, \ldots, \mathbf{V}^{t-m}$ for all $r > m$. If the Markov order of $G_1$ is 1, then all paths from $\mathbf{V}^{t-r}$ to $\mathbf{V}^t$ must be blocked by $\mathbf{V}^{t-1}$ for $r > 1$. Since graphical structure is replicated across timesteps, it follows that all paths from $\mathbf{V}^{t-r}$ to $\mathbf{V}^t$ must be blocked by $\mathbf{V}^{t-u}$ for $r > u$. Therefore, the Markov order of $G_u$ is $u$, which corresponds to Markov order 1 for measurements at rate $u$. □

The following theorem demonstrates correctness of our ASP algorithm.

**Theorem 4.** *Listing 1 is a direct encoding of the undersampling problem.*

*Proof.* We will prove this by contradiction. Let us call the undersampled input graph to the algorithm $\mathcal{H}$, considering that is the undersampled version of a graph $\mathbf{G}_{true}^1$ at rate $u_{true}$. By definition, every directed edge in $\mathcal{H}$ corresponds to a path of length $u_{true}$ in $\mathbf{G}_{true}^1$. Similarly, every bidirected edge in $\mathcal{H}$ corresponds to an unobserved common cause fewer than $u_{true}$ timesteps back(refer to Section 2 for exact definition). Line $7 - 11$ in Listing 1 considers all such $\mathbf{G}^1$s without exclusion. Let us call the set all the pairs of graphs and corresponding undersampling rates $u$ described by Listing 1 $\mathbf{S}$.

Let us assume there is a pair $\mathbf{G}_a^1$ and $u_a$ that is in $\mathbf{S}$ but if we undersample $\mathbf{G}_a^1$ by $u_a$, let us call it $\mathbf{G}_a^u$, will not be the same as $\mathcal{H}$. If $\mathbf{G}_a^u$ has an extra directed(bidirected) edge, this will contradict with line 12(13) of Listing 1. Similarly, if $\mathcal{H}$ has a directed(bidirected) edge that in not present in $\mathbf{G}_a^u$, it will contradict with line 14(16). Therefore, Listing 1 is a direct encoding of the undersampling problem. □

## B   EXAMPLES OF CHANGES IN GRAPHS THROUGH UNDERSAMPLING

In this section, we provide additional examples to visualize how graphs will change through undersampling.

## C   COMPARING SRASL AND A MODIFIED VERSION THE HYTTINEN ET AL. (2017)

As mentioned in Section 2, Hyttinen et al. (2017) specifies the undersampling rate $u$. Therefore, a comparison of their method with ours will not be a fair one. However, one can iterate over undersampling rates to find all the solution at different undersampling rates. In this section, we compare the performance of this modified version of Hyttinen et al. (2017) to our proposed method. Figure 9 summarizes this experiment. As we can see, proposed method in Hyttinen et al. (2017) performs compatibly with our method in small graphs. However, as the graphs grow larger, the advantage of our method gains significance. Specifically, Hyttinen et al. (2017) struggles with large graphs and larger undersampling rates. Most of the test cases on graphs larger than 30 nodes and undersampling greater than 3 did not complete in the dedicated 24-hour period. While our method was able to compute the equivalence class of all the graphs much faster than 24 hours.

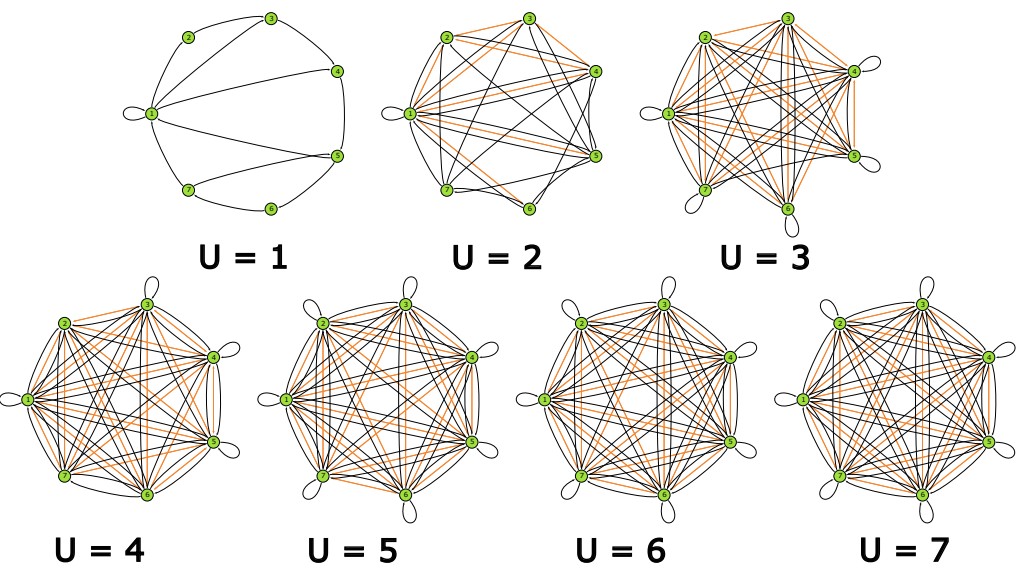

Figure 7: Example of a 7-node graph undersampled 6 time.

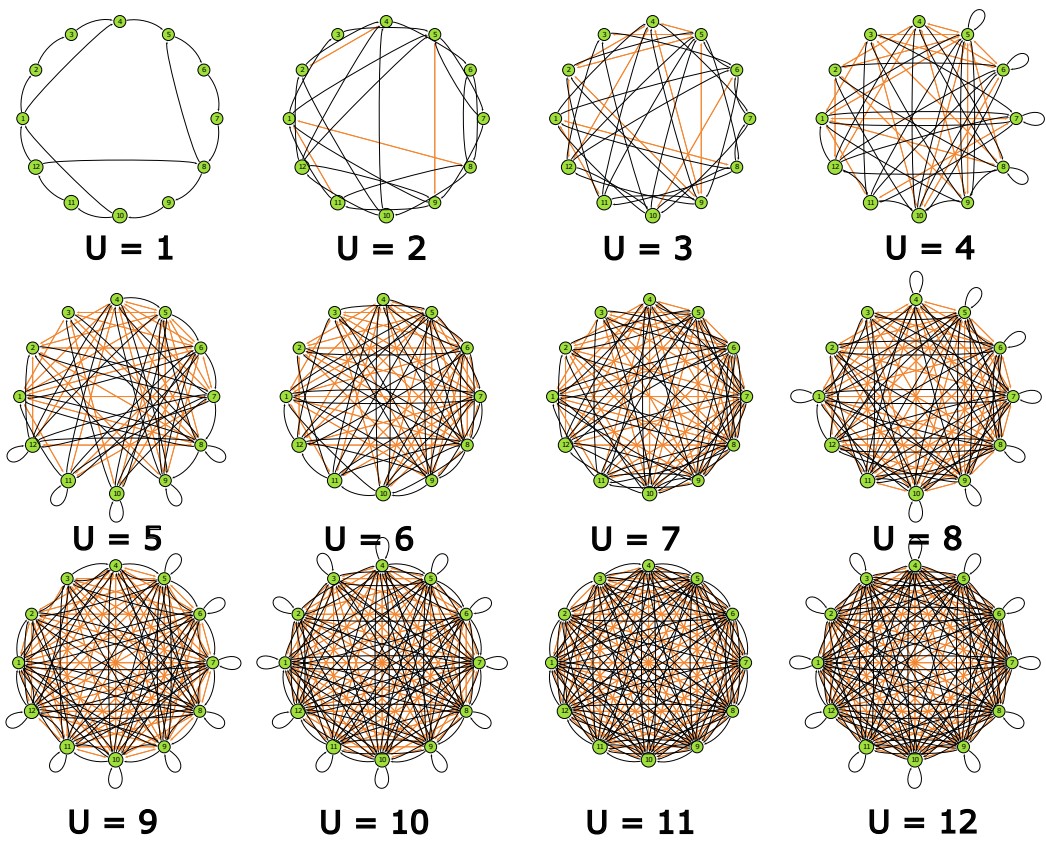

Figure 8: Example of a 12-node graph undersampled 11 time.

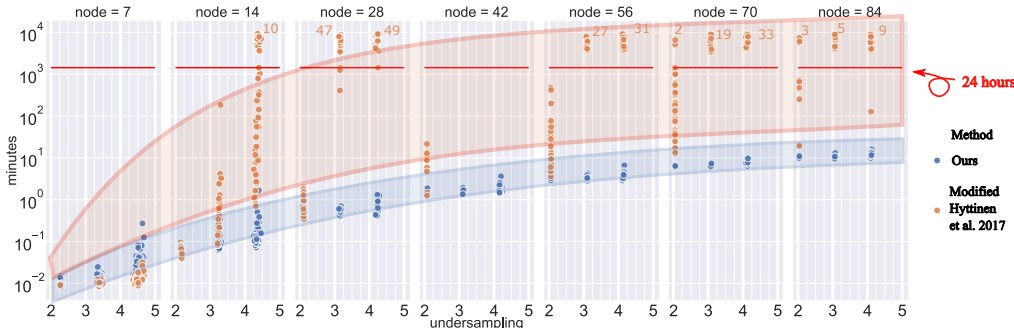

Figure 9: Time behavior of the same set of graphs when solved with our proposed method (blue) and modified version of Hyttinen et al. (2017) with iterating over undersampling rates (orange). The time out for this experiment was 24 hours and the numbers in orange indicate number of examples that was not completed in 24-hour period.

## D  The Effects of Accounting for SCCs In sRASL

In this section, we show the results of additional experiments on the effects of accounting for strongly connected components (SCCs) when the graph has a modular structure (i.e., consists of several interconnected strongly connected components). For this experiment, we generated 50 random graphs sized 8 to 15 with multiple SCCs as described in Table 1. Then on the same set of graphs, we ran sRASL once with using our additional constraints for SCC structures and once without accounting for the modular structure. We limited the computational resources available to each run to 24 hours time cutoff with a RAM limit of 50 GB. The results presented in Figure 11 show that using additional constraints to account for SCC structure dramatically reduces the time and memory needed to compute equivalent classes for undersampled graphs. Furthermore, the difference between time and memory requirements to solve for these graphs with and without constraints for SCCs increases for larger graphs as the computational requirements for the latter grow at a much faster pace. This result allows us to handle much larger graphs as shown in Figure 5 of the main paper.

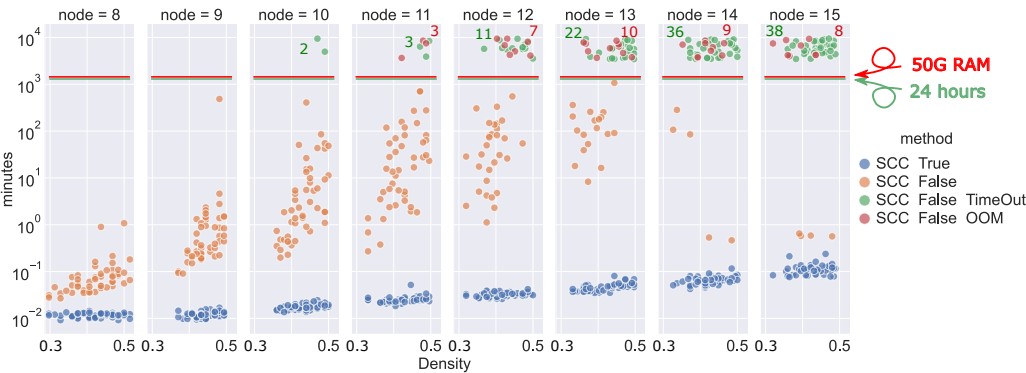

Figure 10: Time behavior of the same set of graphs when solved with and without accounting for additional constraints accounting for the SCC structure. While sRASL most of the 15-node graphs in a 24 hours period without the SCC constraints due to either timeout or Out Of Memory error(OOM), the longest it takes to solve a 15-node graph with SCC constraints is 14 seconds. None of the graphs failed to compute the complete equivalence class within the time and memory allocated when solved accounting for the SCC structure.

Table 1: Number of SCCs and nodes per SCC of the graphs in the benchmark dataset

| Num Nodes | 8 | 9 | 10 | 11 | 12 | 13 | 14 | 15 |
|---|---|---|---|---|---|---|---|---|
| Num SCCs | 2 | 3 | 3 | 3 | 3 | 3 | 3 | 3 |
| SCC Sizes | 4,4 | 3,3,3 | 3,3,4 | 3,4,4 | 4,4,4 | 4,4,5 | 4,5,5 | 5,5,5 |

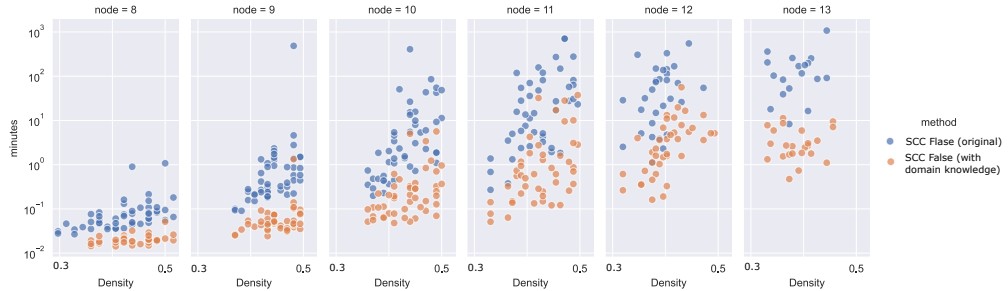

Figure 11: A knowledge of a definite presence of an edge in $\mathcal{G}^1$ between, for example, nodes 3 and 4, i.e. $V_3^t \to V_4^{t+1}$, can be easily encoded by adding ` edge1(3,4).` to Listing 1. In this experiment, we have added knowledge about a pair of arbitrary selected edges of $\mathcal{G}^1$ to the problem specification (orange dots) and compared the run time with the ASP specification that does not include this additional information about the solution (blue dots). The time out for the new computation was set to 1 hours and the examples were all the same as the ones already shown in Figure 1. The speed up with the additional constraints is clearly visible on the plots.

# E    sRASL APPLIED TO REAL fMRI DATA

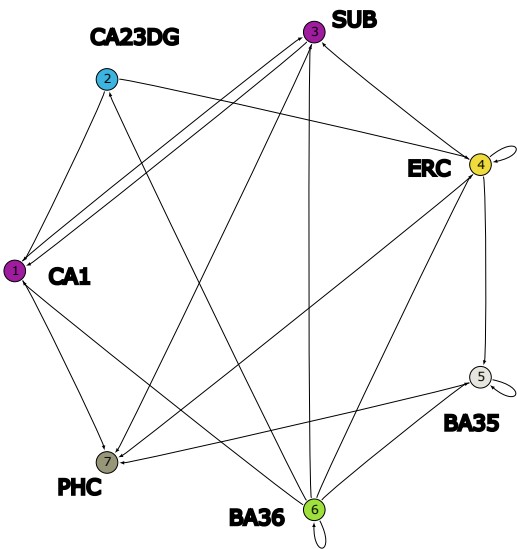

Figure 12: Estimated graph from fMRI data of resting state left hemisphere medial temporal lobe using sRASL after obtaining measurement time scale graph by applying Granger Causality. Regions of interest include cornu ammonis 1 (CA1), and dentate gyrus together (CA23DG); entorhinal cortex (ERC); perirhinal cortex divided in Brodmann areas (BA35 and BA36); and parahippocampal cortex (PHC)..

In order to demonstrate the the application of our method, we used publicly available data from (Sanchez-Romero et al., 2019) and applied our method on the resting-state fMRI data. We used the 10 datasets of concatenated recording for 10 individuals, comprising seven regions of interest from medial temporal lobe, each containing 4,210 datapoints.

We first generated estimated graphs $\mathcal{H}$ from the fMRI data using Granger Causality (Granger, 1969; Cook et al., 2017). Note that these estimated graphs are at the measurement timescale (they include bidirected edges) and due to statistical and measurment error are often not reachable from any graph at causal time scale $\mathbf{G}^1$. Therefore, we apply sRASL optimization on $\mathcal{H}$ to get the closest reachable graph at causal time scale. Figure12 shows the result of our estimated graph at causal time scale. It is important to note that with empirical data, we do not have fully defined ground truth to assess our findings.

Following our approach on Section 4.5, we use the estimated graph $\mathbf{G}_{opt}^1$ in Figure 12 and undersample it by the rate that our sRASL optimization has found, i.e. $u_{opt}$ to get $\mathbf{G}_{opt}^u$. We then use sRASL to obtain $[\![\mathbf{G}_{opt}^u]\!]$ (i.e., the full equivalence class of the undersampled graph that is "nearest" to $\mathcal{H}$). In the case of resting state fMRI data from left hemisphere medial temporal lobe, the full equivalence class consists of 23 graphs that are shown in Figure 13. From this class of equally possible underlying causal graphs, psychologists and experts can examine and determine with causal graph is the most reasonable one.

## F    Brief Introduction on clingo and Answer Set Programming (ASP)

clingo (Gebser et al., 2011) combines a grounder gringo and a solver clasp. clingo is a declarative programming system based on logic programs and their answer sets, used to accelerate solutions of computationally involved combinatorial problems. The grounder converts all parts of a clingo program to "atoms," (grounds the statements) and the solver finds "stable models." In ASP, the answer set is a model in which all the atoms are derived from the program and each "answer" is a stable model where all the atoms are simultaneously true.

A general clingo program includes three main sections, which we show below using our algorithm as an example:

1. **Facts:** these are the known elements of the problem. For example, the input to Listing 1 is a graph for which we know the edges. A directed edge from node 1 to node 5 is in $\mathcal{H}$ translates to *hdirected(1,5)* (line 1) or if node 1 is part of the SCC number 2, we state this fact in clingo by *scc(1,2)* (line 2).

2. **Rules:** much like an if-else statement, a rule in clingo consists of a body and a head, formatted as *head :- body.* If all the literals in the body are true, then the head must also be true. Rules can include variables (starting with capital letters), and they are used to derive new facts after grounding. For example:

$$\text{directed(X, Y, 1) :- edge1(X, Y).} \tag{2}$$

means that for any instantiations of the variables $X$ and $Y$, if we have an edge from $X$ to $Y$, there is a directed path from $X$ to $Y$ of length 1. Before this line, if the model contained the fact *edge1(2,3)*, this line would generate a new fact: *directed(2,3,1)*.

Another type of rule is the "choice rule" that describes all the possible ways to choose which atoms are included in the model. For example, in line 5 of Listing 1 we used a choice rule to state that the undersampling rate *u* can be anything from 1 to *maxu*. The cardinality constraint:

$$\text{\{u(1..20)\}.} \tag{3}$$

will generate $2^{20}$ different models (they will not all actually be generated if they conflict with other predicate in each model, or else it would not be possible). In each of these $2^{20}$ models, one subset of all possible atoms generated with this choice rule exists ($\phi$, {u(1)}, {u(1), u(2)}, ...). An example of an unconstrained choice rule is line 6 in Listing 1, where we want to generate one model for each possible way edges can be present in a graph between two nodes $X$ and $Y$. We can also limit the choice rule. In our problem, only one undersampling rate is present at each solution. We limit the cardinality constraint to have only one member in each model:

$$\text{1 \{u(1..20)\} 1.} \tag{4}$$

the 1 on the left is the minimum instantiations of this atom in the model and the 1 on the right is the maximum. Therefore, we only generate $\binom{20}{1} = 20$ models with this rule, namely one for each undersampling rate. Having several choice rules will multiply the number of generated models by each choice rule.

3. **Integrity Constraints:** if choice rules are to generate new models, integrity constraints are there to remove the wrong models from the answers set. More specifically, an integrity constraint is of the form:

$$\text{:- L0, L1, ...  .} \tag{5}$$

where literals $L_0, L_1, ....$ cannot be simultaneously positive. For example, in line 16 of Listing1, we have:

$$\begin{aligned}\text{:- edge1(X, Y), scc(X, K), scc(Y, L), K != L,} \\ \text{sccsize(L, Z), Z > 1, not dag(K,L).}\end{aligned} \tag{6}$$

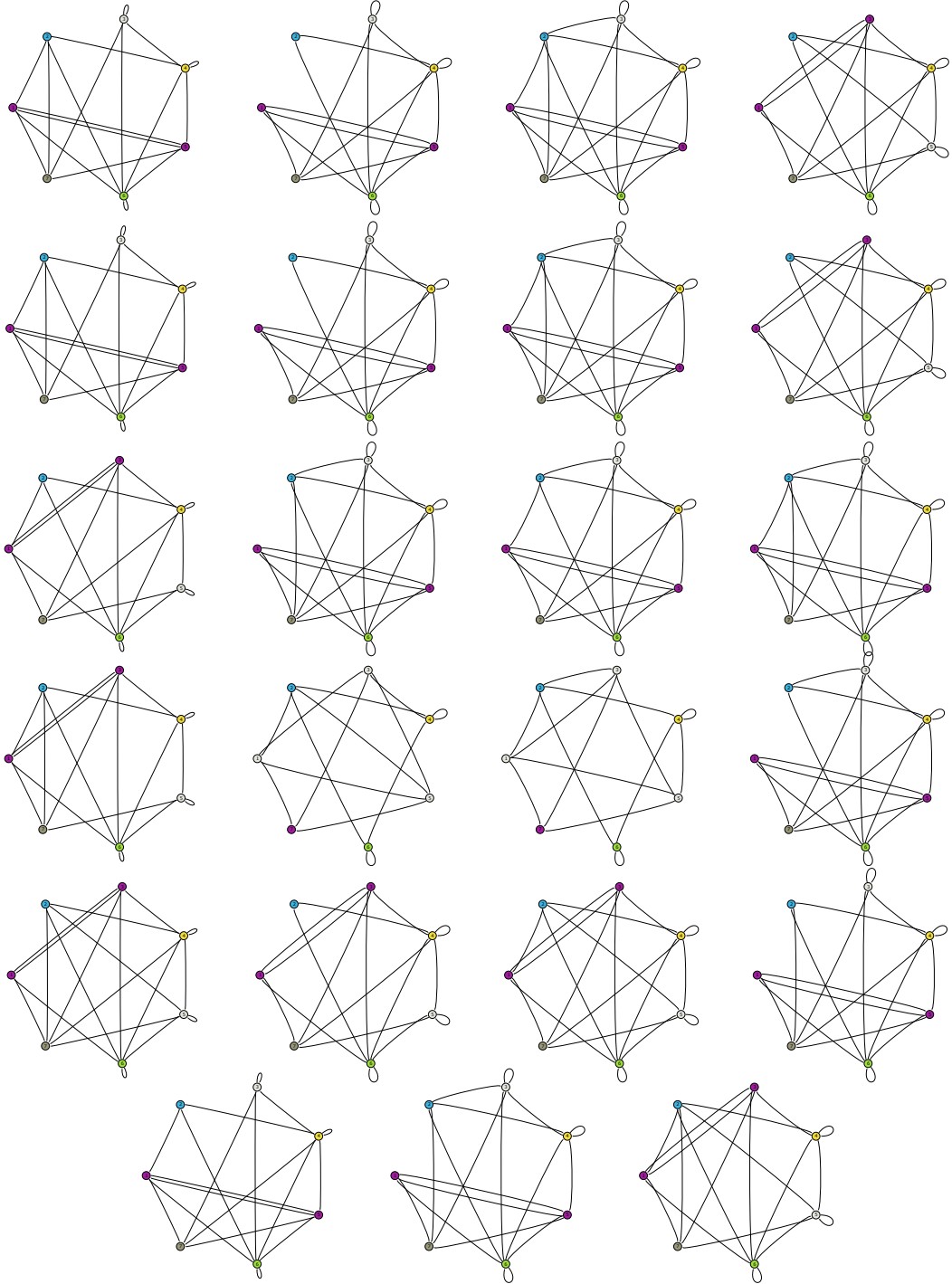

Figure 13: Equivalence class ($[\![\mathbf{G}^u_{opt}]\!]$) of all possible graphs at causal time scale ($\mathbf{G}^1$s) that can be undersampled to and reach $\mathbf{G}^u_{opt}$. Psychologists and experts can examine and determine with causal graph is the most reasonable one

for cases where the graph consists of several SCCs that are connected using a DAG. If the SCCs are connected by a cyclic directed graph, then the whole graph will become one big Strongly Connected Component. Integrity constraint 6 states that if there is not a directed edge from a node in SCC K to a node in SCC L as part of the initial DAG, there cannot be such *edge1(X, Y)* from node X to node Y, if node X is in SCC K and node Y is in SCC L.

