# OpenReview forum: "GRACE-C: Generalized Rate Agnostic Causal Estimation via Constraints"
_ICLR.cc/2023/Conference — ICLR 2023 notable top 25%_

### Official Review · Reviewer_FQ5L · 2022-10-24

**Confidence:** 3
**Correctness:** 4
**Technical Novelty And Significance:** 4
**Empirical Novelty And Significance:** 2
**Recommendation:** 6

**Clarity, Quality, Novelty And Reproducibility:**

The quality seems good, and I do not doubt its originality at all.  It is also very clearly written.

**Strength And Weaknesses:**

UPDATE: Since the authors have added the references and explained the reasoning behind other choices, I will revise my score upwards.

----

A clear strength is the improvement in the speed of discovering these underlying structures, which is impressive.

There are several weaknesses, however.  First, a significant part of the literature on causal discovery in time series is entirely omitted.  A short list (I am sure it is not exhaustive) would include Granger (1969), Chu and Glymour (2008), Entner and Hoyer (2010), Malinsky and Spirtes (2018), Jabbari et al. (2017) and Gerhardus and Runge (2020).  I would also expect to see a comparison of the method to at least one other state-of-the-art discovery algorithm, rather than to a worse incarnation of the method in the paper.

Second, the graph is assumed to be first order Markov without instantaneous effects, and the under-sampling assumed to be at a fixed integer; there is no reason to believe that either of these assumptions will hold in a real problem.

Third, there is no real data application.  This is not necessarily a problem, but given that the motivation focuses very strongly on fMRI data and the paper claims that this new algorithm ought to be able to handle such data, I was disappointed not to see it applied.

### References

Chu, T. and Glymour, C. (2008). Search for Additive Nonlinear Time Series Causal Models. _Journal of Machine Learning Research_, 9:967–991.

Entner, D. and Hoyer, P. (2010). On Causal Discovery from Time Series Data using FCI. In _Proceedings of the 5th
European Workshop on Probabilistic Graphical Models_, pages 121–128.

Gerhardus, A. and Runge, J. High-recall causal discovery for autocorrelated time series with latent confounders. In _Advances in Neural Information Processing Systems_ 33 (2020): 12615-12625.

Granger, C. W. J. (1969). Investigating causal relations by econometric models and
cross-spectral methods. _Econometrica_, 37:424–438.

Jabbari, F., Ramsey, J., Spirtes, P., and Cooper, G. (2017). Discovery of Causal
Models that Contain Latent Variables Through Bayesian Scoring of Independence Constraints. In
_Machine Learning and Knowledge Discovery in Databases_, pages 142–157.

Malinsky, D. and Spirtes, P. (2018). Causal Structure Learning from
Multivariate Time Series in Settings with Unmeasured Confounding. In _Proceedings of 2018 ACM SIGKDD Workshop
on Causal Disocvery_, volume 92, pages 23–47, PMLR.


**Summary Of The Paper:**

This paper provides an Answer Set Programming based algorithm for determining a true directed causal structure from an under-sampled version of it.  The original problem is NP-complete, and previous methods have not been successful with even moderate-sized graphs, but the new method (sRASL) can deal with up to 100 nodes.


**Summary Of The Review:**

UPDATE: Raised score from 5 to 6.

----

I can't give a higher rating than this unless the authors can explain their omission of so much (seemingly) related literature from the paper, and their reasons for only comparing to one (almost a straw-man) method.  If these things can be explained to my satisfaction, then I would be happy to raise my score.

### Minor Comments

1. The comparison at the end of section 2 is slightly unfair, since you take the longest computation from RASL and then give its sRASL counterpart.  You could presumably give the longest computation from sRASL and then say what its RASL counterpart was!

2. Similarly, the comment on page 9 ('except for high undersamplings') sounds wrong, given that you only use a maximum undersampling of 4; this is far lower than the 20 you describe as being likely in fMRI data.

### Typos
 - throughout: many `\cite` (or `\citet`) commands should be replaced by `\citep`.
 - page 1: 'importantly incorrect' - this is a strange choice of phrase.
 - page 2: the references for PAGs and MAGs appear to be the wrong way around.
 - page 4: 'Section4.3' $\to$ 'Section 4.3'.
 - page 5: 'In order to incorporate Equation 5 in Listing 1': this comment comes out of the blue, and no mention of the equations in Appendix C has previously been made.
 - page 9: 'tool in the scientists’ policy-makers’ toolboxes' - is there a missing word here?
 - pages 10-11: please ensure that your references are consistent and that the capitalization is correct.

---

> ### Author Response · Authors · 2022-11-19
> **Discussion on assumptions**
>
> >Some references are not cited
>
> We agree that we should have cited some of the algorithms that can be used to learn the measurement timescale graph ${\cal H}$ that is provided as input to sRASL. We added those in the updated version. We emphasize, though, that these papers are all addressing a different core challenge than sRASL, as they all learn causal structures at the measurement timescale.
>
> All of the mentioned papers can be used to generate the input to sRASL and as such cannot be compared but can be combined with it. We have mentioned this in the updated paper.
>
> >Assumption on first order Markov
>
> We agree that this is a key assumption, but we suggest that it is a relatively benign one, since higher orders can be modeled by adding random variables to function as memory. As proved in the appendix(lemma 2) , undersampling cannot increase Markov order. To account for a higher order Markov structure of the DBN we can encode each edge $V^t_i \rightarrow V^{t+k}_j$, where $k>1$ in additional random variables. E.g. $V^t_1 \rightarrow V^{t+2}_1$ would be encoded by adding $V_\text{new}$ and two edges $V^t_1 \rightarrow V^{t+1}_\text{new}$ and $V^t_\text{new} \rightarrow V^{t+1}_1$. This encoding reuses all previous results.
>
> > without instantaneous effect:
>
> We aimed to address this concern in footnote 1 (page 2) and Lemma 1 in the appendix. Since physical phenomena take time to propagate, there is a time scale that resolves all apparently instantaneous interactions. We only assume that our causal time-scale is no longer than needed to resolve these interactions. Notably, neither the undersampling process, nor latent variables (removal of entire random variables with their univariate time series) are able to introduce instantaneous directed edges. We have demonstrated this claim in the appendix Lemma 1.
>
> >Integer undersampling rate
>
> Since the causal timescale can potentially be relatively fine-grained, we expect that this assumption is less constraining than it might seem. Additionally, phase jitter, temporal noise, and other random effects should spread the notion of exact causal rate, thus a solution to the RASL problem should find a rate that is nearest to the causal timescale in an average sense. That being said, non-integer undersampling is a very interesting possibility that would require careful thought about the underlying causal metaphysics (e.g., what do we mean by a “half-timestep causal influence”?). We look forward to investigating the question in more detail.
>
> >Undersampling lower than 20
>
> We have mentioned this in Footnote #9: Of course, the actual undersample rate could be much lower than 20. Voxels typically contain 8 − 10 layers of neurons, so the “causal timescale of a voxel” could easily be as high as 1000 ms (i.e., u = 2).
>
> >The comparison at the end of section 2 is slightly unfair, since you take the longest computation from RASL and then give its sRASL counterpart.
>
> We agree that this comparison seemed unfair. To resolve this, we also compared the slowest sRASL and compared it with its RASL counterpart. While the slowest graphs for sRASL takes 20.446 Seconds to compute the equivalent class, the same graph takes 780.3 Minutes (13 Hours) for RASL to compute. We have updated the manuscript with this information.
>
> >Typos
>
> Thank you for bring these mistakes to our attention. We have fixed the mentioned errors.

---

> > ### Comment · Reviewer_FQ5L · 2022-11-23
> > **Follow-up**
> >
> > Thank you for adding the references.  I will revise my scores up a little, though it would have been nice to see a comparison to at least one of the other methods, and an application of the new approach.

---

> > > ### Author Response · Authors · 2022-11-23
> > > **Clarification**
> > >
> > > Thank you for accounting for our response and increasing your rating.  Note, following reviewers' suggestions we have compared our method to a baseline method (Hyttinen et al. (2017)) in Appendix Section C of the latest revision of our manuscript.  We have also added an application to fMRI data and found an equivalence class of generating graphs at the causal timescale (Appendix Section E).

---

> > > > ### Comment · Reviewer_FQ5L · 2022-11-24
> > > > **Signposting!**
> > > >
> > > > Thanks for that.  Can you make sure these sections are sign-posted in the paper?  I can't see any reference to Appendix E, and the only reference to Appendix C seems to be that it explains what ASP is.
> > > >
> > > > Also, the title of Appendix C should be "Comparing sRASL and **a** modified version **of** Hyttinen et al . (2017)"

---

> > > > > ### Author Response · Authors · 2022-11-29
> > > > > **Updated**
> > > > >
> > > > > Thank you for your suggestion. We will make sure to sign-post these sections for the final version. We have mentioned these experiments in the main text of the latest version of the manuscript on our end. Please let us know if there's anything else we should do to get the manuscript ready for publication.

---

### Official Review · Reviewer_yZ3C · 2022-10-24

**Confidence:** 3
**Correctness:** 4
**Technical Novelty And Significance:** 2
**Empirical Novelty And Significance:** 2
**Recommendation:** 6

**Clarity, Quality, Novelty And Reproducibility:**

Clarity: The paper is well-written and the various ideas are explained thoroughly, except for a few issues discussed above.

Novelty: The proposed work appears to be marginally novel as it is based on ideas from previous works (shown below).
 - Search-based algorithm for the same task by (Plis et al., 2015).
 - Structural insights about SCCs in a compressed graph by (Danks & Plis, 2013)
 - Using ASP to learn the causal graph when the rate is known by (Hyttinen et al., 2017)

I am open to revisiting this evaluation based on elaboration from the authors.

**Strength And Weaknesses:**

Strengths: The evaluation of the proposed algorithm shows a significant speedup compared to RASL and thus provides a clear advantage in terms of scalability.

Weaknesses & comments:
1- Source of undersampled graph H: The authors state the following:
> sRASL algorithm takes as input a (potentially) undersampled graph H, whether learned from data D, expert domain knowledge, a combination of the two, or some other source.

It is understandable to make such an assumption whenever there are theoretical guarantees that a compressed graph can be asymptotically learned from data. If such a result is known in the literature, it should be made clear in the text. However, it sounds unlikely that such a structure can be fully learnable as is the case with PAGs (Zhang, 2008).

2- Evaluation: There is no comparison of the performance of sRASL and the work of (Hyttinen et al., 2017). I understand that the latter assumes the undersampling rate is known but a comparison is possible by running "the method sequentially for all possible u".

3- Citation confusion: The citations on page 2 regarding the works of MAGs and PAGs are incorrect. The work of (Richardson & Spirtes, 2002) presents the theoretical formulation of ancestral graphs where MAGs are a special case. However, MAGs are not learnable from data. Zhang (2008) builds on this work by Spirtes and others to formulate a complete learning algorithm, the output of which is a PAG. The cited work by Zhang is for inference applied on top of PAGs; a more suitable citation is the one below.

Zhang, J. (2008). On the completeness of orientation rules for causal discovery in the presence of latent confounders and selection bias. Artificial Intelligence, 172(16-17), 1873-1896.

4- Typos:
+ p.4, "As a concreate example of the improvements": concreate  --> concrete
+ p.5, "In order to incorporate Equation 5 in Listing 1": do you mean Equation 1?
+ p.6, "In other words, Theorem 5 implies": Do you mean Thm. 4? Theorem 5 in the appendix is the exact replica of Theorem 4.

**Summary Of The Paper:**

The work addresses the problem of learning a causal graph, or rather the set of compatible graphs in the equivalence class, from time-series data when facing the challenge of undersampling, that is, when the sampling rate does not match the rate of the data-generating process. It assumes the presence of a compressed graph which can, asymptotically, be learned from time-series data and the objective is to identify all the causal graphs that are compatible with the compressed graph at some sampling rate. Thus, the authors propose an algorithm, sRASL, which reformulates a search-based algorithm, RASL by (Plis et al., 2015), along with additional structural insights into a constraint satisfaction problem. Experimental evaluation shows a clear advantage of the proposed algorithm over the search-based approach.

**Summary Of The Review:**

The proposed algorithm provides a clear advantage over previous methods in terms of speed and scalability. This is evident from the experiment though some further expansion of the results would be helpful (comparison to Hyttenin et al. (2017)). However, the method assembles ideas from previous works making the novelty and contribution marginal.

---

> ### Author Response · Authors · 2022-11-19
> **Feasibility and additional comparison to previous works**
>
> > It sounds unlikely that such a structure can be fully learnable as is the case with PAGs (Zhang, 2008).
>
> We have assumed that there are no latent confounders (which is, in the time series setting, a stronger assumption than the “normal” causal sufficiency assumption). Given that assumption, various methods (e.g., SVAR estimation) will return the compressed graph in the limit of infinite data whenever the distribution for the compressed graph falls within the family permitted by the method (e.g., linear Gaussian with simultaneous connections). We agree that matters are significantly more complicated if there can be latent confounders, as learning from data will yield an equivalence class of graphs (typically, one with infinitely many members), not a single compressed graph. (Other complications also result, such as infinite-lag connections between observed factors.)
>
> >There is no comparison of the performance of sRASL and the work of (Hyttinen et al., 2017).
>
> Hyttinen et al. 2017 solve the problem for a known fixed undersampling rate $u$, while the RASL problem specifically frees the user from that assumption. Additionally, the general ASP formulation made it easier to specify the problem structure, such as SCCs, which we productively exploited in this paper or, potentially, measurement timescale graphs for different rates, or other expert knowledge that spans multiple undersampling rates $u$. The intent here is to simultaneously encode the largest possible set of constraints to speed up the search time by limiting the search space and, where possible, minimize the equivalence class.
>
> However, following the reviewers request, we have extended the work of (Hyttinen et al., 2017) by iteratively looping through the possible values of $u$ and ran a comparison with sRASL. Please find the results in Figure 9 of Appendix C.
>
> Please note, due to the extensive amount of time needed for experiment in Appendix C, and the limited time for discussion phase, not all the columns for the modified version of Hyttinen et al. 2017 have 50 samples. The completed experiment will be updated in the next few days.
>
> > Citation confusion and typos
>
> Thank you for bring these mistakes to our attention. We have fixed the mentioned errors.

---

### Official Review · Reviewer_3inH · 2022-10-26

**Confidence:** 4
**Correctness:** 4
**Technical Novelty And Significance:** 3
**Empirical Novelty And Significance:** 3
**Recommendation:** 8

**Clarity, Quality, Novelty And Reproducibility:**

Overall I thought this paper was very well written, easy to follow, and fairly complete. The novelty is somewhat limited since it is directly building off of prior art, but I don't find that a reason for penalization–the work provides a non-trivial improvement that is important for application. My largest complaint with respect to clarity is the description of the algorithms. The authors use Clingo code to describe the algorithm. While I commend them for including code that can be easily reproduced, pseudocode should be preferred so the algorithms can be more easily interpreted by a wider audience.

**Strength And Weaknesses:**

Strengths:
* Compelling task
* Practical and reasonably scalable approach for learning graphs under under sampling
* Compelling experimental evidence

Weaknesses:
* Would be nice to see a more substantive analysis of the behavior of the algorithm when assumptions fail to hold and the algorithm is finding "close" solutions
* Still relatively modest in scale. While 100 nodes is substantially larger than prior art large scale temporal systems can contain substantially more variables.
* As I mention below, it would be nice to have an algorithmic description of the method in a more common format.

**Summary Of The Paper:**

This work studies the problem of learning a causal network from temporal data in the presence of under sampling. In particular, the authors propose a solver based approach for learning a graph from under sampled data, which improves upon prior art significantly in terms of running time. The core of the approach is converting an existing algorithm (RASL) which is score based to a solver based approach.

**Summary Of The Review:**

Overall, I think this paper provides a nice extension to the existing literature. The problem of learning from under sampled time series is nearly ubiquitous in practical applications. I found the proposed method to be well reasoned, clearly presented, and to have good justification via empirical evidence. While I would have preferred to see a more thorough analysis of the behavior of the algorithm under misspecification, I consider this to be a nice addition to the literature.

---

> ### Author Response · Authors · 2022-11-19
> **On assumptions and future works**
>
> >Algorithm behavior under assumption violation
>
> We agree with the value of further empirical study of various failure modes beside the one we already provide in Figure 6. Note, however, that given how flexible our approach is, a study like that would not be easily conclusive. For example, Figure 6 shows that `clingo` optimization tends to produce sparse graphs. Yet, this is merely an artifact of how we set the optimization weights. We could tune the formulation to favor denser solutions, sparser solutions, or anything in between. This flexibility makes it harder to distinguish between a principled failure mode and a failure to properly set hyperparameters. In agreement with the reviewer we leave such study for future work.
>
>
> >100 nodes is still small in some applications.
>
>
> We absolutely agree with this comment and share the reviewer’s views. We do aspire to have approaches able to handle even larger numbers of nodes that sRASLs current 100 within an hour. We hope our paper inspires further development in this direction. In principle, even our current solution can handle single graphs of much larger sizes if one is willing to wait long enough. This wait was prohibitive for a systematic exploration of the space and we did not explore even the larger sizes. However, as Theorem 1 shows, the problem is NP-complete and we are excited to get an solution usable in a much wider context than previously available algorithms capped at around 6 nodes could support.
>
> > Change clingo code to pseudocode.
>
>
> We appreciate the request and apologize for causing the confusion. As we mentioned elsewhere in this rebuttal, rate agnostic structure learning is a problem, not an algorithm: find an equivalence class containing graphs that match ${\cal H}$ at some different undersampling rate. While RASL of Plis et al. 2015 is an imperative algorithm, our Listing 1 and Listing 2 are problem encoding for the solver. We have modified the captions of these listings and adjusted the text throughout accordingly to minimize the probability of misinterpretation.

---

### Official Review · Reviewer_9RAD · 2022-10-26

**Confidence:** 2
**Correctness:** 3
**Technical Novelty And Significance:** 3
**Empirical Novelty And Significance:** 3
**Recommendation:** 6

**Clarity, Quality, Novelty And Reproducibility:**

The study problem in this paper is quite interesting. The proposed method in this paper is novel and efficient.

**Strength And Weaknesses:**

Strengths:
1. The study problem in this paper is quite interesting, it focuses on learning causal graph from dynamic systems. This problem is quite challenge when the data are undersampled.
2. The proposed algorithm has faster performance without asking for specific undersampling rate.

Weaknesses:
1.  It might be better to compare with more other baselines.
2.  It might be more clear to show how the graph changes with the time changing.

Additional questions:
1.  What is the main difference between sRASL and RASL?
2.  How would the undersampling rate influences the causal discovery results?

**Summary Of The Paper:**

This paper focused on the causal discovery problem on the dynamic systems. The proposed method focuses on the problem that the measurements timescale and the causal timescale do not match. To solve this problem, this paper proposes Rate-Agnostic Structure Learning, that uses undersamples graph H as input learned from data D, domain knowledge or other sources. The learned graph from first order markov shows effective undersampling rate, from the full space of G1. sRASL proposed in this paper does not suffer from assumptions or computational complexities.

**Summary Of The Review:**

In summary, this paper proposes to solve an interesting and challenging problem. The proposed method is sound and valid across different settings of datasets.

---

> ### Author Response · Authors · 2022-11-19
> **Main contributions and some clarifications**
>
> >The main difference between sRASL and RASL
>
> There are three main differences:
> First and foremost, we treat RASL as a problem, not as an algorithm, while the original RASL publication conflates the two. We prepended “s” to distinguish them. This problem is encoded in a declarative language based on the constraints we define in  Listing 1, unlike the imperative implementation of the RASL algorithm. Note, that sRASL in Listings 1 and 2 is only a problem formulation, while the algorithm is decided and carried through by the solver, encoded by “s” in sRASL.
>
> The second difference is that the declarative formulation simplifies adding additional constraints based on derived or expert information. Most prominently, the  advantage we got by encoding the SCC structure, and thereby gaining significant speed-up. This difference is important since it is not immediately obvious how to account for SCC structure in the imperative RASL algorithm. We have performed additional experiments, where presence of some edges is known a priori and encoded into the formulation. This alone without encoding the SCC structure also sped up the solver. (Figure 11 in appendix)
>
> Last, but not the least, sRASL provides a straightforward way to find approximate solutions when ${\cal H}$ is an unreachable graph.
>
> >Undersampling rate influences on causal discovery results
>
> At any undersampling rate, our method finds the complete set of all possible ${\cal G}^1$s such that there is a $u$ where ${\cal H} = {\cal G}^u$ (cf. the earlier discussion of completeness). Thus, the undersampling rate does not influence the correctness of the causal discovery process. However, undersampling rate does influence the number of graphs in the equivalence class of ${\cal H}$: in general, higher undersample rates mean more information is lost, and so the equivalence class of ${\cal H}$ will be larger.
>
> >How graphs change with undersampling
>
> Apologies for lack of complete clarity. We originally believed that a single example in Figure 1 provides an idea of graph change due to undersampling. However, after this comment we see the value of further examples and we have added all undersampled versions of a couple of graphs to the Appendix B, Figures7&8.

---

### Official Review · Reviewer_kYJa · 2022-11-01

**Confidence:** 4
**Correctness:** 4
**Technical Novelty And Significance:** 3
**Empirical Novelty And Significance:** 2
**Recommendation:** 6

**Clarity, Quality, Novelty And Reproducibility:**

The main idea of the paper seems quite straightforward given the previous work on this problem, so the novelty is limited. However, the improvement achieved by the new proposal is significant. The paper is very clear and readable, and I expect the reproducibility of the empirical results to be very good, especially since they do not involve data.

A clarificatory question: does the soundness of the algorithm rely on the assumption that the condition on gcd(L_s) in Theorem 3 (and Theorem 2) holds?

**Strength And Weaknesses:**

Strengths:

1. The proposed algorithm addresses a commonly encountered challenge to causal discovery and achieves a very significant speedup in comparison to previous attempts.
2. The experiments demonstrate some nice properties of the new algorithm.
3. This paper is well written and relatively easy to follow.

Weaknesses:

1. There is no experiment with real or even simulated data. I am a little puzzled why no empirical demonstration is attempted on streamlining the proposed algorithm with a causal discovery algorithm applied to data at the measurement time scale.
2. For the optimization version of the algorithm, it is unclear how the weights should be determined. Are they supposed to be entirely user or expert specified? Or is there a data-driven procedure to assign the weights. Conceivably some causal discovery methods may yield interpretable weights on the inferred edges, but this prospect does not seem to be discussed in the paper.
3. It is also unclear to me why no empirical comparison is made to the ASP-based algorithm in Hyttinen et al. (2017).

**Summary Of The Paper:**

This paper proposes a new algorithm to solve the problem of recovering, as much as possible, the original graphical causal structure at the causal timescale from the derived graphical structure at a measurement timescale, where measurements are made every u number of time steps for an unknown u. The new algorithm is based on ASP and incorporates some new constraints that were not exploited in previous algorithms. Experiments show that the new algorithm scales much better than the previous methods.

**Summary Of The Review:**

This is a well written paper presenting an improved method to tackle an interesting problem. The improvement demonstrated by experiments is significant, though more experiments on simulated or real data would probably better vindicate the utility of the method.

---

> ### Author Response · Authors · 2022-11-19
> **Clarification on weights assignment and more experiment on fMRI data**
>
> >How do we set the weights for the optimization experiment?
>
> The weights may encode the strength of connection. For example, if ${\cal H}$ is estimated as an SVAR model, then the edge-weights may enable `clingo` to preferentially ignore the edges with weaker connection strength (if there is no ${\cal G}^1$ that implies the observed graph). In addition to using observed data to estimate the weights, prior knowledge can play a key role: edges known to exist can be given a higher weight, while those known to not exist could be given reduced (or zero) weight. The approach is flexible in that it can combine estimates from data, and expert knowledge.
> >Does soundness require gcd =1?
>
> No, the soundness of sRASL requires only that Listing 1 encodes the relevant satisfiability conditions, regardless of the gcd of ${\cal G}^1$ or even whether ${\cal G}^1$ is a strongly connected component. If $\operatorname{gcd} > 1$, though, then sRASL will be much more computationally complex because we would not be able to take advantage of the SCC structure.
> >There is no comparison with Hyttinen et al. 2017.
>
> Hyttinen et al. 2017 solve the problem for a known fixed undersampling rate $u$, while the RASL problem specifically frees the user from that assumption. Additionally, the general ASP formulation made it easier to specify the problem structure, such as SCCs, which we productively exploited in this paper or, potentially, measurement timescale graphs for different rates, or other expert knowledge that spans multiple undersampling rates $u$. The intent here is to simultaneously encode the largest possible set of constraints to speed up the search time by limiting the search space and, where possible, minimize the equivalence class.
>
> However, following the reviewer's request, we have extended the the work of Hyttinen et al. 2017 by iteratively looping through the possible values of $u$ and ran a comparison with sRASL. Please find the results in Figure 9 of Appendix C.
> >More experiments on fMRI data and comparison with Hyttinen et al. 2017
>
> We have provided more experiments on the appendix section. Please check the latest version of the appendix for more experiments. Appendix C compares our method with the modified version of Hyttinen et al. 2017, and Appendix E provides additional empirical experiment on fMRI data. Please note, due to the extensive amount of time needed for experiment in Appendix C, and the limited time for discussion phase, not all the columns for the modified version of Hyttinen et al. 2017 have 50 samples. The completed experiment will be updated in the next few days.

---

### Author Response · Authors · 2022-11-19
**To all reviewers**

We thank the reviewers for their time, valuable comments, and recognition of the value of our contributions. Below, we addressed questions and concerns of each reviewer separately. Where appropriate we have changed, updated, and augmented the manuscript following reviewers’ recommendations. The new manuscript is uploaded.

All reviewers requested more clarity on the novelty of the work.

The currently published work on structure learning from undersampled time series is highly abstract and impractical. The parametric methods can hardly handle systems with more than 2 variables, while the nonparametric approaches extend to 6, which is also small. The toy size of the addressable problems prevents most if not all practical uses of the published approaches in turn depriving the theorist and method developers of material and inspiration for further works. That may explain lack of progress on the topic.

Our work demonstrates how by using ASP to  account for modularity in the problem structure (common in natural domains) and achieve speed ups of 4 orders of magnitude  thus enabling work with large practically relevant sized graphs. We believe in this case it is not just a little “faster” but rather crosses a threshold from curious theoretical development to a tool that may finally be used by practitioners. As for the theoretical development, our solution is, importantly, not an approximation but a correct and complete solution to the problem, which we show in Theorem 4. With these contributions we aim to attract due attention from the theory and practitioners groups attending and reading ICLR works.

---

### Decision · Program_Chairs · 2023-01-20

**Decision:**

Accept: notable-top-25%

**Justification For Why Not Higher Score:**

Topic/scope of the paper might not be fully attractive to the general ICLR audience.

**Justification For Why Not Lower Score:**

Clear consensus that this is relevant work and well executed.

**Metareview: Summary, Strengths And Weaknesses:**

Interesting causal structure discovery paper taking in consideration the issue with under sampling. A key point is that measurements might not happen exactly when they are "wanted" (for causal inference). They propose an approach using ASP that improves on SOTA, in particular in terms of speed up, which is significant. The paper is clearly written and has enough original material.


**Note From Pc:**

if the above contains the word "oral" or "spotlight" please see: "oral" presentation means -> notable-top-5% and "spotlight" means -> notable-top-25%. As stated in our emails, we are disassociating presentation type from AC recommendations